# The Splendors and Miseries of Open Access Scientific Publishing in Ukraine

Andriy Novikov [1,2] 

[1]  Department of Biosystematics and Evolution, State Museum of Natural History NAS of Ukraine, Teatralna str. 18, 79008 Lviv, Ukraine; novikoffav@gmail.com
[2]  Department of Botany, Biology Faculty, Ivan Franko National University of Lviv, Hrushevskoho str. 4, 79004 Lviv, Ukraine

**Abstract:** The letter represents the author's opinion on the reasons and background of the actively developing practice of unconscientious open access scientific publishing, as well as briefly discussing the current condition of academic publishing and indexation in Ukraine.

**Keywords:** open access; academic publishing; Ukraine

Open access (OA) to scientific publications has always been the traditional option in Ukraine, which was also stated in the "Belgorod Declaration on open access to scientific knowledge and cultural heritage" in 2008 [1]. Most academic journals in Ukraine are non-commercial and fully subsidized by hosting institutions. They have always been focused on the distribution of printed copies and usually grant free access to electronic versions of the published materials [2,3]. However, many Ukrainian academic journals did not, for a long time, have websites, nor did they update their contents in it when they had one. Only in 2018 did the Ukrainian government introduce a new law with a mandatory requirement for scientific publishers to regularly update websites and to provide DOIs (Digital Object Identifiers) for all published content [4]. This resulted in faster growth of Ukrainian scientific publications with free access.

In total, there are 3029 scientific journals registered in Ukraine [5,6]. However, only 10.00% of these journals have passed open licensing and are currently included in the Directory of Open Access Journals (DOAJ) database [7]. Only 3.47% of Ukrainian scientific journals are indexed by Clarivate Analytics [8], and only 1.75% by Scopus [9]. Moreover, most published content does not correspond to FAIR principles [10,11] due to issues with provided metadata. There are no special investigations on the implementation of FAIR principles by Ukrainian publishers yet. However, even a quick browsing through 20 randomly selected journals shows that only one of them fully adheres to the FAIR principles. Another 11 journals have various issues with provided metadata (i.e., incomplete or incorrect metadata, lack or absence of metadata on parental web-pages and/or in pdf versions of the articles, absence of separate web-pages for each article). Additionally, eight journals from this subset have no online presence whatsoever or have considerable issues with provided metadata and content. Despite free access to published materials, in many cases, Ukrainian scientific publications are poorly indexed by search engines (i.e., Google Search Console) because of missing or incorrectly provided metadata.

Another problem is that many Ukrainian investigations are conducted at a low quality level [12] due to both the low level of financial support and the lack of proper control. Researchers in Ukraine are not encouraged to publish in international high-rated editions because this usually does not give them either extra points nor bonuses. Only in the past few years, some of the Ukrainian universities have begun to pay bonuses for publishing in high-rated journals. However, these are local rather than global initiatives. Moreover, Ukraine has a national list of qualified scientific journals (NLQSJ),

which is more important for Ukrainian scientists. Publications in journals included in the NLQSJ are often of higher value. NLQSJ contains only Ukrainian academic journals and is subdivided into three categories [3–5]. The main category A includes 78 Ukrainian journals indexed by Web of Science or Scopus. The other two categories, B and C, cover journals indexed by any international scientometric database. This means that even if the journal is indexed by a predatory database (e.g., Open Academic Journal Index), it can be included in these categories. Hence, many Ukrainian academic journals strive for inclusion in such databases instead of improving the quality of their publications.

Publications in A-rated journals still do not give any real advantages in Ukraine and are not required either for dissertation defense or employment [13,14]. Therefore, many Ukrainian scientists continue to publish their outcomes in low-quality journals, and often the only ones who cite these articles are themselves. In some disciplines, self-citation of Ukrainian scientific publications reaches 72%, which makes Ukraine one of the leading countries in the world in this ranking [15].

Basing on my experience as a scientific editor for the last eight years, I can assume that the problem of quality publications in Ukraine has a complex character and cannot be clearly separated from scientific practice. I believe that every Ukrainian academic journal today struggles with a problem that has no simple solution. If editors of Ukrainian journals decide to reject low-quality manuscripts, they are faced with an absence of authors who can provide publications of an acceptable level. Vice versa, if editors accept such low-quality manuscripts, they doom their journal to never becoming better. In such situations, to close the journal would be the simplest and fairest solution. However, taking into account that most Ukrainian academic journals are maintained by research institutions [16], the closure of academic journals becomes an internal institutional and state political problem [17]. Some journals are trying to reform gradually, introducing slightly more and stricter requirements. However, in many cases, they face opposition or even sabotage from authors, when the authors intentionally do not follow the rules and recommendations of editors and reviewers in their manuscripts. As a result, journals often have a dilemma: to accept material as it is or to miss releases of entire volumes. Thus, many Ukrainian authors are hindering the reform of Ukrainian academic journals.

The government of Ukraine is also trying to find some ways to resolve this situation. In particular, in 2016, the Ukrainian government wanted to tighten the requirements for scientific publications, but this initiative was met with protests, especially from the side of Ukrainian humanities academics, who believe that national science should co-exist independently of the international scientific community. The Ukrainian humanities academics argue their position by stressing a few main issues: (a) the impossibility of developing common rules for the evaluation of scientific activities in different disciplines, (b) the inability to compete with colleagues from abroad, (c) the willingness to protect peculiarities of national, traditional scientific schools, (d) the impossibility of publishing their outcomes in high-rating editions [17–20].

Of course, the existence of 105 Ukrainian journals indexed by the Web of Science and 53 journals listed in the SJR database is hopeful. Nevertheless, many Ukrainian scientific publications, even if found, produce scientific noise rather than enhance the world's scientific heritage [21–24]. Unclear copyright politics and lack of anti-plagiarism verification of manuscripts supplement this situation [25]. High levels of self-citation, plagiarism (including self-plagiarism and translations from other languages, in particular, from Russian), and the unfair practice of data compilation/falsification, together with lack of qualified peer-review, make many of the Ukrainian scientific publications untrustworthy.

It is fair to say that in the last decade, many efforts have been made to improve the practice of open access scientific publishing in Europe and Ukraine. In particular, launching the OpenAIRE initiative in 2009, with the introduction of the Zenodo repository in 2013, made a breakthrough in access to many scientific publications, including research papers, datasets, and other supporting materials [26,27]. Today, Zenodo is actively used by many Ukrainian scientists for self-archiving. Open Journal System (OJS), which provides a free and reliable tool for the management of open access journals, is also often applied by Ukrainian academic publishers. The "Open Science in Ukraine" initiative provides free consulting support for Ukrainian academic journals in their development on the basis of OJS [28].

The Open Ukrainian Citation Index (OUCI) launched in late 2019 provides a convenient search tool for open scientific publications [29]. OUCI should become the first step in the development of a national scientometric index in Ukraine, which was preliminarily introduced several years ago [21]. However, Ukraine is still far from other European countries in its open access scientific activities. For example, Ukraine occupied the last position in the list of countries that have obtained grants covering article processing charges in the framework of European FP7 projects [27,30]. De-Castro and Franck [27] concluded that this "is the result of a very complex mix of cultural and socio-economic factors and the reasons behind it would be worth a deeper investigation". I totally agree with De-Castro and Franck, and I do hope that this will be a question for forthcoming deeper investigations.

**Funding:** This research received no external funding.

**Conflicts of Interest:** The author declares no conflict of interest.

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
