# Peer review of "The Splendors and Miseries of Open Access Scientific Publishing in Ukraine"

_publications, doi:10.3390/publications8010016_

Round 1

Reviewer 1 Report

The text appropriately describes the issues that the current Ukrainian publishing landscape faces. Being a letter and not an article, not too much of an improvement can perhaps be expected in terms of the structure, but the fact that there is no mention to the OpenAIRE networks and its widespread initiatives to support Open Access publishing is both a major shortcoming and a missed opportunity for some constructive criticism.

The author seems fully unaware of the fact that the coordination for the OpenAIRE network in Central Eastern Europe is in fact in Kyiv, not to mention a number of initiatives to promote and support sound technical Open Access publishing via APC-free Gold Open Access journals. Some references are provided below for a number of these efforts to strengthen the Open Access publishing landscape :

  • "Funded bids for the Alternative Funding Mechanism for APC-free Open Access journals and platforms", https://www.openaire.eu/blogs/funded-bids-for-the-alternative-funding-mechanism-for-apc-free-open-access-journals-and-platforms-1
  • Funding APCs from the research funder’s seat: Findings from the EC FP7 Post-Grant Open Access Pilot
    (section 1.7 The alternative funding mechanism for APC-free Open Access journals and platforms)
    https://doi.org/10.3145/epi.2019.jul.13 
  • OpenAIRE alternative funding for non-author fee publishing platforms: IBL PAN (Poland)
    https://www.openaire.eu/blogs/openaire-alternative-funding-for-non-author-fee-publishing-platforms-ibl-pan 
  • Kotilava – Finnish academic journals towards immediate Open Access
    http://kotilava.fi/19-elokuu-2016-1247/kotilava-%E2%80%93-finnish-academic-journals-towards-immediate-open-access 
  • Norwegian open journals in the social sciences and humanities
    ["A key element is that publishing in these journals should not require payment of an APC"]
    https://www.openaccess.no/english/humsam/ 

An additional section summarising some of these initiatives (esp those in the region) and highlighting the advisability of seeing some similar effort in Ukraine to the one taking place in most European countries would much improve the text (see the section "Country highlights" in the OpenAIRE blog at https://www.openaire.eu/blogs/categories/countryhighlights)

Author Response

Thank you for efforts and valuable comments!

I have completed my manuscript with additional paragraph describing some of initiatives focused on developing of scientific open access in Europe and Ukraine. I also included into the References some of proposed publications. I did not included in my paper citations for Polish, Finnish, and Norwegian initiatives that you listed because I believe that comparison of different national OA initiatives can be an independent and interesting investigation, which was not the aim of this letter of opinion. Moreover, comparison of different national OA initiatives requires deeper analysis in context of national socio-political and economical environments. I feel that I am not ready to do such investigations at this moment.

I did not find any information about coordination of OpenAIRE in Kyiv. I have checked the recent list of OpenAIRE's National Open Access Desks (NOADs) and did not locate any such information: https://www.openaire.eu/contact-noads ; https://www.openaire.eu/frontpage/country-pages Perhaps, it was coordinated before, but, unfortunately, I cannot find such information too.

Reviewer 2 Report

This is an opinion paper and does not require the same standards as regular research paper.

However, the paper should be improved before acceptation, in three ways:

Improvement of English style and language. Especially the meaning of the 3rd section is difficult to understand.

Adding of sources and figures to support the expressed opinion: for the first section on Ukranian journal publishing; more details on the number of UA journals (disciplines...); on the journals "FAIR status" (2nd section).

Clear distinction between at least two different problems: the scientific practice and integrity of UA academic authors, and the characteristics of UA journals (quality, selection, business model...). 

Why do UA authors not need to publish in international journals (line 34)? 

What is the link between "predatory database" and the UA research policy (line 39)?

I can't get the message of the letter: "Do we really need open access to such publications?" So what: improve the quality of UA journal publishing (how)? Reject more bad manuscripts (how)? Reduce the number of OA journals? Why should this be a specific UA problem? 

Author Response

Dear Reviewer,

Thank you for efforts and valuable comments!

Reviewer (R). This is an opinion paper and does not require the same standards as regular research paper.

Author (A). Yes, it is my pilot paper in this field. And it is rather representing my opinion. Hopefully, it will be a background for further deeper investigations.

R. However, the paper should be improved before acceptation, in three ways:

Improvement of English style and language. Especially the meaning of the 3rd section is difficult to understand.

Adding of sources and figures to support the expressed opinion: for the first section on Ukranian journal publishing; more details on the number of UA journals (disciplines...); on the journals "FAIR status" (2nd section).

Clear distinction between at least two different problems: the scientific practice and integrity of UA academic authors, and the characteristics of UA journals (quality, selection, business model...).

A. I improved most of the article and my colleague helped me with English. Section 3 has been almost totally rewritten. Article also was completed with additional last section following request of another reviewer. I also updated data and recalculated percents accordingly to the new information from databases. I completed the manuscript with more details on FAIR and provided additional references. However, I did not make analysis by disciplines, because there is no such information available.

Basing on my experience as a scientific editor for last eight years, I believe that problem of quality publications in Ukraine has complex character. Problem of quality of Ukrainian scientific publications cannot be clearly separated from scientific practice. Because, even if Ukrainian journal rejects low quality manuscripts, it faces with issue of absence of authors, who can provide acceptable level of publications. And from opposite, if Ukrainian journal accepts low quality publications, it will never become better. The only solution is not to publish at all. Taking into account that most of Ukrainian academic journals are running by research institutions, the closing of the journals become internal political issue. Ukrainian government tries to find some solution for this situation. Some of Ukrainian journals also try to reform slightly implementing more and stricter requirements to manuscripts. But in many cases journals are faced with opposition from the side of authors or even sabotage, when authors do not follow the rules and do not want to accept recommendations of editors/reviewers. In such cases journals have choice to publish materials 'as is' or miss the publication of the full volume. I do not analyze business models of all Ukrainian academic journals and also I do not pay so much attention to procedure of selection of academic journals. However, basing on my experience of work in two different Ukrainian journals, I can surely say that quality of publication in Ukraine definitely depends both from authors and publishers that work in tandem. Anyway, this paper is just an opinion letter that does not pretend on deep investigation with statistics' support and comprehensive analysis.

R. Why do UA authors not need to publish in international journals (line 34)?

A. I corrected this in text. Authors do not need to publish in international journals because this usually does not provide any benefits for them, neither in career nor in salary.

R. What is the link between "predatory database" and the UA research policy (line 39)?

A. I corrected this in text. The link is next: Ukrainian journals must to be indexed in the international databases to be included in the Ministry’s list of qualified journals. However, Ministry does not indicate which exactly databases are qualified. Therefore many of Ukrainian journals prefer to index in predatory databases than to do hard work for inclusion in WoS or Scopus. This also partly depends from the absence of ‘good authors’, who can provide good citable publications.

R. I can't get the message of the letter: "Do we really need open access to such publications?" So what: improve the quality of UA journal publishing (how)? Reject more bad manuscripts (how)? Reduce the number of OA journals? Why should this be a specific UA problem?

A. Yes, you are right, this message was rhetorical. It also made a negative sense for manuscript. Therefore I excluded it.

Round 2

Reviewer 1 Report

Thanks for sharing this updated manuscript plus the link to the author's reply to the original review. I do indeed believe the manuscript has now been significantly improved and that the sole barrier currently standing between its current state and its release in MDPI Publications is the language style.    I am attaching a few suggestions to fix the most relevant style issues -- these updates should leave the text ready for publication, although the author is encouraged to have the text proof-read by some native English speaker before its final submission if possible.

Author Response

Dear Reviewer,

Thank you for detailed corrections that you proposed and general positive evaluation!

I implemented all proposed stylistic and grammatical corrections in my manuscript; this really helped to make it better readable. After that I also sent it to professional proofreader, who introduced several other corrections. The only thing, which I did not accept is the writing of the word “humanities” from a capital letter. In context of my manuscript, the word “humanities” has general meaning and hence, I believe, has not to be capitalized. The proofreader also did not propose to start “humanities” from caps. However, if this is necessary, I would be thankful if editors correct it.

Moreover, I added one more paragraph on request of second reviewer. This paragraph has supplementing character and originated from my previous personal response to Reviewer 2.

Reviewer 2 Report

The revised version is better, compared to the first paper. I think there are still some mistakes with spelling and grammar but this should be assessed by an English native speaker. 

I like the author's personal statement which is part of his response to the reviewer:

"Basing on my experience as a scientific editor for last eight years, I believe that problem of quality publications in Ukraine has complex character. Problem of quality of Ukrainian scientific publications cannot be clearly separated from scientific practice. Because, even if Ukrainian journal rejects low quality manuscripts, it faces with issue of absence of authors, who can provide acceptable level of publications. And from opposite, if Ukrainian journal accepts low quality publications, it will never become better. The only solution is not to publish at all. Taking into account that most of Ukrainian academic journals are running by research institutions, the closing of the journals become internal political issue. Ukrainian government tries to find some solution for this situation. Some of Ukrainian journals also try to reform slightly implementing more and stricter requirements to manuscripts. But in many cases journals are faced with opposition from the side of authors or even sabotage, when authors do not follow the rules and do not want to accept recommendations of editors/reviewers. In such cases journals have choice to publish materials 'as is' or miss the publication of the full volume."  

Perhaps consider to integrate it in some way or other into the opinion paper.

Author Response

Dear Reviewer,

Thank you for valuable suggestion, I really appreciate your efforts on making my manuscript better!

I added one additional paragraph describing influence of authors on scientific publishing in Ukraine, just as you asked. I slightly modified this paragraph following suggestions of proofreader and also completed it with two additional references.

As I mentioned before, I sent my manuscript to professional proofreader who made a number of stylistic and grammatical corrections in the text. Moreover, I implemented several stylistic corrections proposed by Reviewer 1. I hope that now my manuscript has an acceptable level.